# Construction of Asbestos Slate Deep-Learning Training-Data Model Based on Drone Images

**DOI:** 10.3390/s23198021

**Published:** 2023-09-22

**Authors:** Seung-Chan Baek, Kwang-Hyun Lee, In-Ho Kim, Dong-Min Seo, Kiyong Park

**Affiliations:** 1Department of Architecture, Kyungil University, Gyeongsan 38428, Republic of Korea; seungchan1318@gmail.com (S.-C.B.); kh.lee@kiu.kr (K.-H.L.); 2Department of Civil Engineering, Kunsan National University, Kunsan 54150, Republic of Korea; inho.kim@kunsan.ac.kr; 3School of Architecture, Civil, Environmental and Energy Engineering, Kyungpook National University, Daegu 41566, Republic of Korea; dmseo@knu.ac.kr; 4Department of Big Data, Chungbuk National University, Cheongju 28644, Republic of Korea

**Keywords:** deep learning, drone survey, asbestos slate, image classification

## Abstract

The detection of asbestos roof slate by drone is necessary to avoid the safety risks and costs associated with visual inspection. Moreover, the use of deep-learning models increases the speed as well as reduces the cost of analyzing the images provided by the drone. In this study, we developed a comprehensive learning model using supervised and unsupervised classification techniques for the accurate classification of roof slate. We ensured the accuracy of our model using a low altitude of 100 m, which led to a ground sampling distance of 3 cm/pixel. Furthermore, we ensured that the model was comprehensive by including images captured under a variety of light and meteorological conditions and from a variety of angles. After applying the two classification methods to develop the learning dataset and employing the as-developed model for classification, 12 images were misclassified out of 475. Visual inspection and an adjustment of the classification system were performed, and the model was updated to precisely classify all 475 images. These results show that supervised and unsupervised classification can be used together to improve the accuracy of a deep-learning model for the detection of asbestos roof slate.

## 1. Introduction

Asbestos is utilized in various industrial applications due to its thermal resistance, strength, and insulation. However, since the 1960s, asbestos has been known to cause serious risks to human health and numerous countries have instituted bans or strict regulations [1,2]. Asbestos and its microscopic fibrils can be released into the atmosphere and are easily inhaled or ingested; when asbestos enters the body, these microscopic fibrils can cause serious health problems such as the development of asbestosis pulmonum, lung cancer, mesothelioma, and other cancers [3,4,5].

Before the risks associated with asbestos were identified, asbestos was utilized widely across various industries in South Korea, similar to the case of many other countries. In South Korea, 96% of imported asbestos was used for manufacturing slate in the 1970s and 82% in the 1990s [6,7,8]. Asbestos was predominantly utilized as a building material. However, in 2009, after the risks of using asbestos slate were discovered, South Korea banned the production, import, and utilization of asbestos and products containing asbestos [9]. Asbestos slate refers to roofing materials containing asbestos fibers. It was extensively used before its ban owing to its durability and fire resistance. However, if asbestos slate is not removed, fibers can be released into the air, creating a hazardous environment. If significant quantities of asbestos are detected in asbestos slate roofs, appropriate measures should be adopted to minimize exposure and ensure the safety of individuals working with, or in the surroundings of, asbestos. In South Korea, 1.2 million pieces of asbestos slate have been used, which corresponds to approximately 18% of all buildings [10,11]. According to Kim et al. [12], the use of asbestos slate is projected to cause 555 deaths by 2031 in Korea. In this regard, the safe removal of asbestos slate is a high priority.

Asbestos-containing products should be dismantled or demolished in accordance with the ban on their use, and the procedures for the visual investigation of materials with asbestos, such as tex installed indoors, are relatively simple to perform. However, the visual examination of existing asbestos slate that has been neglected for a long time is difficult because it is installed on the roof. Specifically, assistive tools such as scaffolds and ladders are necessary for this purpose, which increases the time, labor, and the overall cost of the procedure [13]. Because a field survey is essential for removing asbestos slate used on roofs, a safer and more efficient method than visual examination is required.

To address this issue, drones have been employed to survey roofs for asbestos slate. Surveys of asbestos slate via drones are effective and safe, and are efficient alternatives to other dangerous, laborious, and high-cost methods. A recent study explored asbestos slate on roofs, utilizing the spatial characteristics of drone flight. Zhang et al. [14] targeted housing redevelopment districts and conducted a comparative analysis of existing methods of surveying asbestos slate and drone-based methods to confirm the usefulness of drones in field surveys. Even though the drone-based survey method has the advantages of rapid survey, economic feasibility, and less labor intensiveness, this method still has low credibility and objectiveness, as drone images must be interpreted by humans. To supplement these weaknesses, object detection techniques using deep learning, which are accurate and reliable, have been widely applied according to the recent development of object recognition techniques. Seo et al. [15] targeted a residential area where asbestos slate was widely used, and utilized deep learning, which can automatically detect asbestos. Comparative verification with the traditional visual inspection method enabled the rapid detection of asbestos slate roofs with high accuracy. However, errors occurred due to the small size of the learning dataset.

By applying deep learning after constructing labeled datasets to automatically detect and identify asbestos slate based on visual pattern images, the accuracy of asbestos slate detection can be increased. However, a large amount of learning data is required in some cases, and, as a result, a methodology for constructing learning datasets is required. The construction of learning datasets is an important process for developing object detection models with accuracy and reliability via deep learning. Creating a high-quality dataset necessitates multiple rounds of data collection, pre-processing, and evaluation. During this phase, the application of both supervised and unsupervised classification techniques has the potential to generate more comprehensive and accurate learning datasets. Supervised learning ensures the creation of a model to accurately classify the asbestos slate, whereas unsupervised learning helps with data exploration and potential data enrichment.

As for asbestos slate detection, the construction of learning data models encompasses the following processes: collection of a large set of images with asbestos slate, labeling of the valley troughs and ridges of the slate, and the annotation of images to indicate the location of the slate in the image. It is necessary to analyze the learning data and train the AI to recognize the unique functions and features of asbestos slate; the more diverse and comprehensive the learning data, the easier it is for a deep-learning model to detect asbestos slate in various situations. The construction of a learning dataset provides the deep-learning model with the necessary information to accurately detect asbestos slate, avoiding the occurrence of false positives and false negatives, and, consequently, detection failure. 

Therefore, in this study, we constructed a learning dataset via an image classification technique to detect asbestos slates using deep learning. To ensure the accuracy and reliability of the deep-learning model for detecting asbestos slate, high-quality training datasets were constructed. By constructing learning data models via image classification techniques, we aimed to create a deep-learning model that can detect objects by automatically classifying asbestos slate images into several categories according to visual characteristics. In this regard, we targeted a specific region in which asbestos slate roofs are widely utilized, acquired an aerial photograph, and attempted to create a learning data model after extracting the shape information of the asbestos slate via supervised and unsupervised classification techniques while targeting the aerial images.

## 2. Materials and Methods

### 2.1. Construction of Learning Dataset Model

In this study, we utilized aerial images to obtain pattern images of asbestos slate via supervised and unsupervised classification techniques. High-resolution aerial images provide a higher level of spatial detail than low-resolution images, and, therefore, can significantly improve the identification and distinction of different objects. Highly specific functions can be utilized in the image classification process. Furthermore, as high-quality images contain more spectral information, it is possible to classify the images more accurately and distinguish Class information when obtaining pattern images of asbestos slate.

Therefore, when acquiring aerial images, we set the altitude at a level that allows for the visual identification of valley troughs and non-valley troughs of the asbestos slate. We obtained aerial images while excluding images where some asbestos slate could not be visually identified. We also employed the auto-pilot flight method by setting an altitude, while targeting buildings with asbestos slate roofs. An overview of the experiment of learning data construction methodology can be found in Figure 1.

In the aerial photography stage (a) of Figure 1, high-resolution images were obtained by low-altitude flight, and an area with many buildings with asbestos slate roofs was selected as the research target area. Before operating and filming with an unmanned aerial vehicle (UAV), permission was sought for flight and filming. In step (b), unsupervised classification was performed based on the obtained aerial images, and the valley trough and ridge (non-valley trough) of the asbestos slate were classified as Spectral Class. The asbestos slate pattern images were extracted by overlapping the valley trough and ridge Classes of the unsupervised classified asbestos slate images. In step (c), we constructed asbestos slate pattern images extracted by unsupervised classification as training data, and, finally, we conducted supervised classification of asbestos slate images on one image.

### 2.2. Data Collection and Acquisition

When constructing asbestos slate deep-learning data using supervised and unsupervised classification techniques based on drone-based images, it is crucial to secure the visibility of images between valley troughs of asbestos slate for accurate classification. Specifically, both altitude and ground sampling distance (GSD) significantly affect the desired level of visibility. Drone images should be high quality with high resolution and sharpness, which is essential for capturing and identifying roof details, including asbestos slate, during classification. To obtain high-resolution images that capture the space between the valley troughs of the asbestos slate in detail, drones should be flown at a low altitude. This will allow for the accurate detection and classification of the space between the valley troughs. GSD refers to the physical distance between two consecutive pixel centers measured on the ground, and it determines the level of detail and resolution of captured images. The lower the GSD, the higher the visibility between the valley troughs of the asbestos slate. Therefore, as each pixel in an image represents a smaller physical realm on the ground, higher-resolution images are created. Altitude and GSD both have an impact on the visibility of asbestos slate; therefore, striking a balance between the two is important. 

If an altitude is set that is too low, images can overlap and undergo distortion, and the flight may be time-consuming. If an altitude is set that is too high, the visibility may be poor and the resolution too low. Therefore, it is necessary to ensure a GSD value that can ensure visibility between the valley troughs of asbestos slate, by setting the altitude in line with the surrounding environment, camera, and drone specifications. The gap between valley troughs of asbestos slate was 6 cm, and, in this study, we obtained drone-based images by setting an altitude of 100 m to satisfy the minimum GSD value of 3 cm/pixel.

When constructing asbestos slate deep-learning data models using supervised and unsupervised classifications based on drone-based images, there are a few important conditions and considerations to ensure high data quality and efficiency. Capturing a variety of samples is important for the creation of deep-learning datasets that can be generalized. The datasets should encompass various lighting conditions, angles, meteorological conditions, and changes in roof conditions, which can appropriately reflect real-world scenarios. For supervised classification, accurate ground truth labels were manually added to drone-based images, and areas corresponding to asbestos slate should be marked to create labeled datasets for training a deep-learning model. As imbalanced datasets can lead to a biased model and affect the classification accuracy, the ratio of asbestos slate samples to non-asbestos slate samples was adjusted to ensure balanced datasets.

For unsupervised classification, an unsupervised clustering algorithm was applied to drone-based images to identify and group similar areas, which can potentially show the existence of an asbestos slate. It can be useful for initial data exploration and to understand the structure of a dataset. When using drone-based images, it is essential to comply with privacy laws and regulations. Therefore, it was necessary to confirm whether all personal or sensitive information was appropriately anonymized and whether drones were flown in accordance with regional regulations and permits.

### 2.3. Image Classification Techniques

Image classification is a method of grouping elements with similar spectral features to help users easily understand images, by analyzing the spectral features of numerous pixels composing an image. Each group is known as a Class, and the technique is applied to extract information that can be classified into several types or steps, such as rice paddies and fields in the target area, from images. The technique can be divided into supervised and unsupervised classification methods based on pixels [16]. 

For specific image classification with the purpose of constructing deep-learning datasets, the data can be classified into unsupervised or supervised based on the methods of arranging patterns with similar features into identical Classes. Unsupervised classification is performed using only image information without a training set, whereas supervised classification is conducted after selecting a certain Class, and then a training set; in this study, we utilized a maximum-likelihood classification algorithm [17,18]. 

Unlike in unsupervised classification, sample patterns belonging to each Class are utilized in supervised classification. In the latter, sample patterns belonging to each Class are utilized in the training/learning step in order to find representative characteristics of the Class. An unclassified and random pattern is assigned to one of the Classes using representative features [19].

As it is impossible to predict patterns belonging to a Class in an unsupervised classification, information about each Class cannot be extracted. Therefore, the images should be classified into categories according to their features. This process is called clustering, and clusters are classified based on features after extracting characteristics from each cluster. Clustering can be conducted to create a sample pattern for supervised classification when a sample pattern is not given in advance or is unknown. Supervised classification has the advantage of high accuracy, and the weakness of struggling to apply Classes with a small distribution rate. Unsupervised classification is advantageous in that it does not require a training set, and it is easy to obtain information about objects at an early stage of classification, although outcomes can vary considerably depending on assumptions. 

Remote sensing satellite data are also widely utilized in image classification research. Danneels et al. [20] researched a methodology of automatically detecting landslides through remote sensing data by utilizing a supervised classification technique. Lee et al. [21] employed single satellite data and a national land cover map to extract training data, and, subsequently, suggested a method for classifying land cover. To this end, initial training data models were constructed by utilizing ISODATA and land cover maps, and classification accuracy was improved after the Maximum Likelihood Classification (MLC) training data were utilized for the land cover classification of satellite images, and Markov Random Fields (MRFs) were applied for each repeated phase to reduce salt and pepper noise [17,22]. As such, supervised and unsupervised classifications are generally applied to satellite image analysis, but drone-based images have a higher spatial resolution than satellite images. This high resolution enables a more detailed analysis, and thus easier detection and better distinction of asbestos slate on the roof. In addition, drone-based data have unique features such as flexible data collection and higher data quality, and, due to these features, this image classification technique is extremely useful in various applications, including the detection of asbestos slate.

Specifically, when considering the configurational properties of asbestos slate, the utilization of an image classification methodology and the construction of an image label through the patterning of shapes between the valley troughs of asbestos slate could prove to be useful. This methodology may effectively aid in generating learning data for the purpose of detecting asbestos slate through deep-learning techniques.

### 2.4. Experimental Design

When conducting an experiment to develop a learning-data construction methodology, it is necessary to first select an appropriate drone satisfying the requirements for aerial photography. This study utilized Inspire2, a UAV with a Zenmuse X5s camera that can capture high-resolution images with sufficient details to accurately identify asbestos slate roofs, after considering factors as the flight stability, camera quality, payload capacity, and battery life. The Zenmuse X5s camera has a Micro Four Thirds sensor with 20.8 MP resolution and has excellent flexibility in terms of the focal distance and viewing angles. Table 1 and Table 2 list the Inspire2 model and specifications of the Zenmuse X5s camera. 

We targeted an area comprising numerous buildings with asbestos slate roofs (Figure 2) and obtained a total of 475 asbestos slate aerial images from this area. To ensure full coverage and sufficient image resolution, area size and shape, flight altitude, and overlapping image capture were considered, and the flight planning software or applications were applied to automate flight paths and optimize image capture.

We confirmed whether the drone captured images at a resolution with a clear and detailed visualization of the asbestos slate. High-resolution images can provide better-quality data for analysis and identification. However, it was important to adjust the resolution balance with consideration to the storage capacity and processing requirements. The image quality was also improved by adjusting camera settings such as exposure, white balance, and image format. As the camera settings can influence the perceived color and texture of asbestos slate roofs, we obtained optimal image quality by collecting images using various settings and evaluating the results. 

By considering these factors, it was possible to collect high-quality image data for constructing asbestos slate learning data, by effectively utilizing drone-based aerial images. We also complied with safety and legal regulations during the shooting.

## 3. Results

### 3.1. Experiment Results 

We applied ArcGIS software to use the patterned images between the valley troughs of asbestos slate in the target drone-based aerial images as training data for supervised classification. 

In previous studies, in which large areas were targeted and unsupervised classification techniques were applied to land cover and satellite images, classification was smoothly achieved only after setting the number of Classes to 10 or fewer as the spatial resolution was extremely poor. However, in this study, the drone-based aerial images had a high spatial resolution, but specific classifications were required due to various components in an area where there are concrete, asphalt, and various roof shapes with different colors and patterns in surrounding environments. However, indiscriminate Class classification negatively affects accuracy during the analysis stage. Therefore, we considered the characteristics of the surrounding environment of the target images, attempted the categorization of several Classes, and ended up with 250 Classes. Among the unsupervised classification methods, we utilized Iso Cluster Unsupervised Classification, which is the most widely used method due to its simple operation and minimal requirements for human intervention. The Iso Cluster tool enables unsupervised classification based on patterns and spectral properties within images. When necessary, we also integrated these with other data layers such as region of interest, function extraction, and statistic computation. Figure 3 shows the unsupervised classification of asbestos slate images. 

Among the 250 unsupervised classified images, we reclassified images based on the RGB values of the pixels between the valley troughs of the asbestos slate. Therefore, it was possible to extract asbestos slate images with external patterning characteristics. Figure 4 shows the reclassified images.

We utilized the unsupervised classified asbestos slate images to construct training data for supervised classification. We distinguished asbestos slate images from non-asbestos slate images to specifically classify asbestos slate images in the target area, and constructed training data for supervised classification.

After constructing the training data with classified patterns of asbestos slate via the unsupervised technique, we utilized the supervised classification technique to extract asbestos slate images from classified patterns. For this process, it was necessary to label the data to communicate which pixels in the classified pattern corresponded to asbestos slate and which did not. Therefore, it is necessary to capture the features distinguishing asbestos slate from non-slate. In general, texture, color histograms, or shape attributes can be included. In the learning stage, labeled learning data were utilized, and the maximum-likelihood classification method, which is a supervised classification algorithm, was applied. This algorithm is based on particular extracted Classes (slate or non-slate). In the classification phase, all drone-based images based on the trained maximum-likelihood classification are utilized, each pixel feature is evaluated, and Class labels are assigned based on the most probable assumptions during the classification process. Figure 5 shows the combined result of these phases. 

### 3.2. Verification of Learning Data 

We conducted a field survey to verify deep-learning training data based on supervised and unsupervised classifications. The survey aimed to secure the reliability of classified asbestos slate images using the constructed deep-learning training images, and to determine the status of discrepancy. 

Additionally, we visited the area with asbestos slate roofs within the research target area and visually examined roof materials in each target place to confirm the presence or absence of asbestos slate roofs. We compared the classification results from supervised classification with measurement data collected in the field, as well as compared the classification per target place with the presence and absence of asbestos slate roofs that were observed in the field. 

As a result of field verification, we identified 12 errors among a total of 475 asbestos images. The errors were attributed to Class misclassification, and, subsequently, we adjusted the Class parameters and reclassified them to reduce the error. As a result, 475 asbestos slate images were precisely classified, and we determined that the data could be utilized as learning data. 

Verification in the field enabled us to evaluate the accuracy and reliability of the map classification process for asbestos slate images, and it is expected that errors or discrepancies can be identified and resolved, ultimately leading to better classification results and higher efficiency.

## 4. Discussion

The rapidly growing field of deep learning has revolutionized various applications across industries. However, it relies heavily on the availability of high-quality labeled datasets. Manual data annotation is time-consuming and costly, which hinders the widespread application of deep-learning techniques. To solve this problem, a new approach for data construction that involves combining supervised and unsupervised classification techniques in the context of deep learning was proposed.

In particular, because of the unique and distinct configurational property of valley troughs and ridges, the integration of supervised and unsupervised classification techniques was useful for constructing deep-learning training data. It was thought that the strength of one approach could compensate for the weakness of the other approach, leading to more comprehensive and powerful datasets with better performance for classifying asbestos slate images, and a more reliable deep-learning model.

Moreover, already damaged asbestos slates can have a huge impact on the accuracy of a deep-learning model. Although it is possible to detect roofs through the learning process in datasets, damage to asbestos slates hindered the ability of the model to comprehensively detect asbestos slate. Therefore, more diverse and advanced learning data are required. Additionally, the accuracy and reliability could be increased if physical inspection or expert evaluations were incorporated. In general, the construction of asbestos slate deep-learning training data requires careful adherence to safety protocols and Class imbalances; if supervised or unsupervised classifications are fully utilized, comprehensive datasets can be created, which can contribute to the generation of an effective and accurate model to detect asbestos slates.

## 5. Conclusions

As the usage of asbestos is banned because of its health risks, existing asbestos roof slates must be detected and destroyed. In this regard, asbestos slates are often identified via drone-based images and the automatic classification and detection of images via deep learning, rather than via traditional methods with their numerous limitations, such as visual inspection. In this study, we utilized an image classification method to construct deep-learning training data to detect asbestos slates using drone-based aerial images. This involved the unsupervised classification of asbestos slate images according to the patterned configurational properties. The images were classified into 250 Classes via unsupervised classification and were then reclassified based on the RGB values of the pixels between the valley troughs of the asbestos slates. The constructed training data were utilized, and the maximum-likelihood classification, which is the supervised classification algorithm, was applied. Field verification was performed, and the data were used to construct an asbestos slate deep-learning training-data model. During field verification, we identified 12 misclassifications of asbestos slate image data and reduced the number of errors through Class parameter adjustment and reclassification, followed by re-verification. Finally, an asbestos slate deep-learning training-data model was established.

The model is expected to require minimal time and effort to apply compared to traditional inspection methods, which require investigations of wide areas using registered building data or visual inspection. Additionally, as conventional asbestos inspections often require scaffolds or high work platforms, they can be costly and time-consuming, whereas drone inspection is cost-effective, as access to roofs is not required.

## Figures and Tables

**Figure 1 sensors-23-08021-f001:**
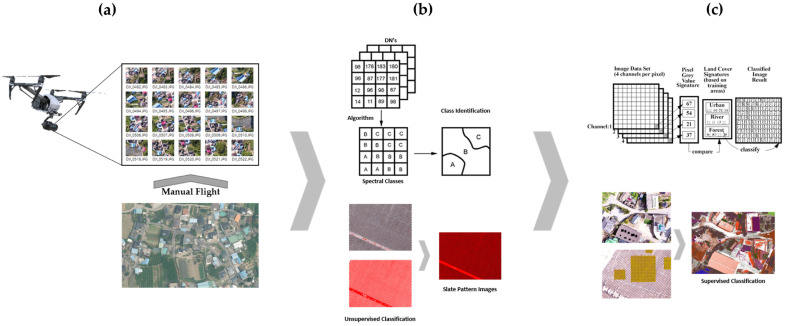
Flowchart of learning data construction methodology.

**Figure 2 sensors-23-08021-f002:**
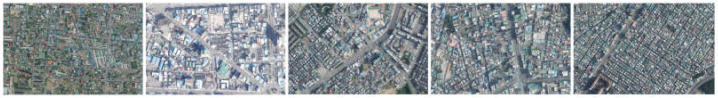
Location of acquired training data.

**Figure 3 sensors-23-08021-f003:**
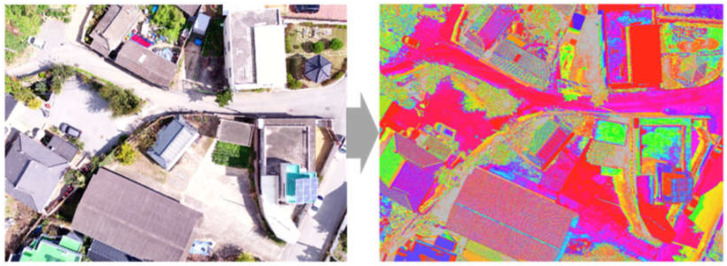
Results of unsupervised classification.

**Figure 4 sensors-23-08021-f004:**
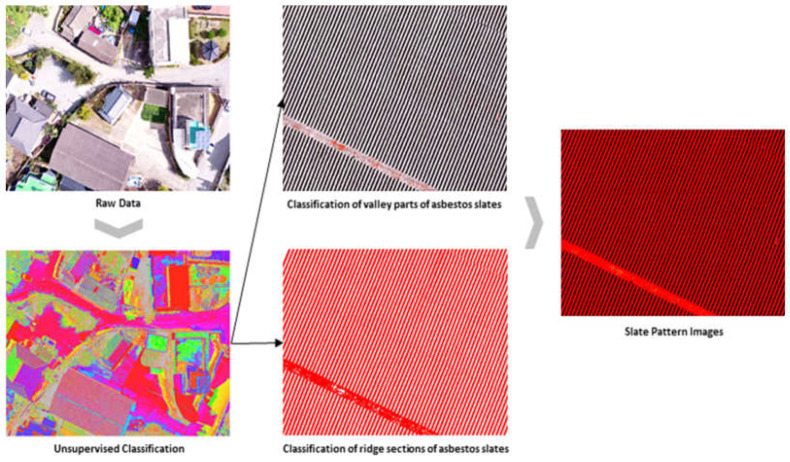
Extraction of asbestos slate pattern image through Class re-classification.

**Figure 5 sensors-23-08021-f005:**
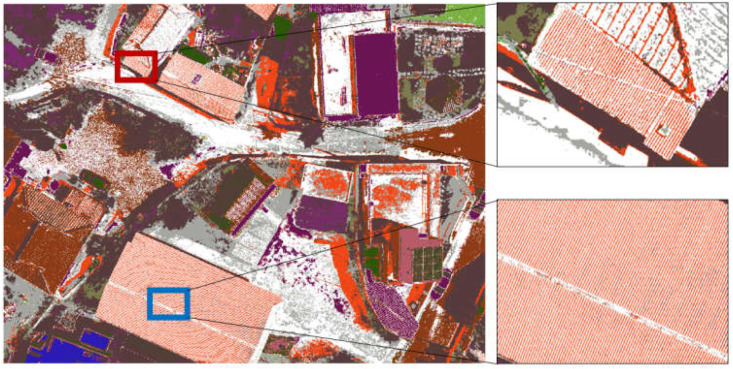
Supervised classification result.

**Table 1 sensors-23-08021-t001:** Specifications of the drone model used.

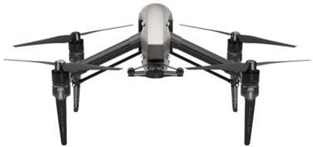	
Classification	Content
Model	Inspire 2(DJI)
Weight	3440 g
Operating temperature	−20~40°
Range of the sensor	30 m
Range of control	7 km
Maximum flight time	27 min
Maximum flight speed	94 km/h

**Table 2 sensors-23-08021-t002:** Specifications of the camera model used.

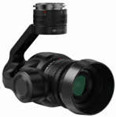	
Classification	Content
Model	Zenmuse X5s
Size	140 × 98 × 132 mm
Operating temperature	−20~40°
Weight	461 g
Effective pixels	20.8 MP
Field of vision	72°

## Data Availability

Not applicable.

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
