# Peer review of "Construction of Asbestos Slate Deep-Learning Training-Data Model Based on Drone Images"

_sensors, 2023, doi:10.3390/s23198021_

Round 1

Reviewer 1 Report

The paper describes a machine learning method to detect and classify asbestos slate roofs using UAV images and a novel approach that combines unsupervised classification to populate a supervised classification sample set in order to create a large training set.  The authors show initial results of correctly classifying 463 out oy 475 images correctly using this method, then improving classification of 475 out of 475 images when ground truth was confirmed, showing the effectivelness of this method.

As this type of roofing seems to be unique to Korea, I wonder if there are roofing materials made of non-asbestos materials in use in Korea that look like the traditional roofs (for architechtural reasons)  that could be mis-classified?  For example, in the United States tile roofs are traditional in certain regions but due to cost and weight, roofing material that mimics the appearance of tile is used but is made of a composite material with the same shape and color as the natural materials.

Reviewer 2 Report

The study is very interesting and very promising. However, it was not successfully presented. After a rich introduction, the sections are poorly explained and not connected.

It is unknown why the authors decided to do a pixel-based classification.

The results must be supported with more figures, accuracy analysis etc.

I suggest adding a flowchart in order to see the steeps the authors used. 

Reviewer 3 Report

The introduction illustrates very well the assumptions and objectives of the work, the subject of which is very interesting, both for the methodology addressed and for the purposes aimed at safeguarding public health. The instruments used and the methodology carried out are well presented and the results achieved are very promising.

The English is understandable and the bibliographic references pertinent.

Publication in the current version is recommended.
